# Saturated Fatty Acids Promote GDF15 Expression in Human Macrophages through the PERK/eIF2/CHOP Signaling Pathway

**DOI:** 10.3390/nu12123771

**Published:** 2020-12-08

**Authors:** Laurent L’homme, Benan Pelin Sermikli, Bart Staels, Jacques Piette, Sylvie Legrand-Poels, David Dombrowicz

**Affiliations:** 1Inserm, CHU Lille, Institut Pasteur de Lille, U1011-EGID, University of Lille, F-59000 Lille, France; Laurent.Lhomme@alumni.uliege.be (L.L.); Benan-Pelin.Sermikli@inserm.fr (B.P.S.); Bart.Staels@pasteur-lille.fr (B.S.); 2Laboratory of Virology and Immunology, University of Liege, GIGA-Signal Transduction, 4000 Liege, Belgium; jpiette@uliege.be

**Keywords:** GDF15, macrophage, obesity, saturated fatty acids, stearate, ER stress, CHOP

## Abstract

Growth differentiation factor-15 (GDF-15) and its receptor GFRAL are both involved in the development of obesity and insulin resistance. Plasmatic GDF-15 level increases with obesity and is positively associated with disease progression. Despite macrophages have been recently suggested as a key source of GDF-15 in obesity, little is known about the regulation of GDF-15 in these cells. In the present work, we sought for potential pathophysiological activators of *GDF15* expression in human macrophages and identified saturated fatty acids (SFAs) as strong inducers of *GDF15* expression and secretion. SFAs increase *GDF15* expression through the induction of an ER stress and the activation of the PERK/eIF2/CHOP signaling pathway in both PMA-differentiated THP-1 cells and in primary monocyte-derived macrophages. The transcription factor CHOP directly binds to the *GDF15* promoter region and regulates *GDF15* expression. Unlike SFAs, unsaturated fatty acids do not promote *GDF15* expression and rather inhibit both SFA-induced *GDF15* expression and ER stress. These results suggest that free fatty acids may be involved in the control of GDF-15 and provide new molecular insights about how diet and lipid metabolism may regulate the development of obesity and T2D.

## 1. Introduction

Growth differentiation factor-15 (GDF-15) is a cytokine belonging to the transforming growth factor-beta (TGF-β) superfamily and plasmatic level is positively associated with the development of several diseases including obesity [1,2], type 2 diabetes (T2D) [3,4], non-alcoholic steatohepatitis (NASH) [5], cardiovascular diseases [6] and various cancers [7,8]. In the past decades, GDF-15 has been extensively studied in cancer where it is mainly regulated by the transcription factor p53 and plays roles in apoptosis, proliferation, invasion, angiogenesis, metastasis, drug resistance and anorexia/cachexia [9,10,11]. More recently, a growing interest about its role in metabolic diseases has emerged, particularly in obesity and T2D. Indeed, similar to its role in cancer-induced anorexia [9], GDF-15 also has an anorectic effect in obesity models. Transgenic mice overexpressing *Gdf15* or mice injected with recombinant GDF-15 are protected against obesity and insulin resistance (IR) [12,13,14,15,16,17]. On the contrary, *Gdf15*-deficient mice are prone to obesity and IR [18].

Glial-derived neurotrophic factor receptor alpha-like (GFRAL) was recently identified as the GDF-15 receptor [14,17,19,20]. The discovery of GFRAL and its high expression within the brainstem area postrema and the nucleus tractus solitaries [14,17,19,20] confirm earlier findings showing that GDF-15 activates neurons in these regions controlling food intake [9,21]. Similar to *Gdf15*-deficient mice, mice lacking *Gfral* are prone to develop more severe obesity and IR [17,19]. Besides the impact of GDF-15 on food intake contributing to the protection against obesity and T2D, anorectic-independent effects of GDF-15 on insulin sensitivity, lipolysis and fatty acid oxidation have been described [12,13,19], suggesting that GDF-15 effects extend beyond the regulation of food intake. 

GDF-15 is expressed by various cell types including adipocytes [22], hepatocytes [23], myocytes [13] and epithelial cells [24]. GDF-15 was originally named macrophage inhibitory cytokine 1 (MIC-1) due to its ability to regulate macrophage activation [25]. GDF-15 is also produced by macrophages themselves [15,25], suggesting that GDF-15 has an autocrine regulatory function. Macrophages are key components of the innate immunity not only playing important roles during infections and cancer but also in tissue development and metabolic homeostasis [26]. Macrophages colonize all the tissues early during embryological development and represent the most abundant immune cell type in most tissues [27]. Tissue macrophage infiltration is also observed in most pathological conditions such as obesity, T2D and cancer [26], indicating that macrophages may represent an important source of GDF-15 in these disorders. To support this hypothesis, overexpression of *Gdf15* in monocytes/macrophages reduces body weight and IR [16]. By contrast, reconstitution of macrophage-depleted mice with *Gdf15*-deficient macrophages increases body weight and IR compared to mice reconstituted with *Gdf15*-proficient macrophages [15], showing that GDF-15 production by macrophages participates in obesity and T2D. Despite these findings demonstrating a role of macrophage-produced GDF-15 in obesity and T2D, little is known about the stimulus and the molecular mechanism driving *GDF15* expression in macrophages in metabolic diseases.

One major pathway regulating GDF-15 in myocytes, hepatocytes and cancer cells is the endoplasmic reticulum (ER) stress [13,23,28,29]. Importantly, ER stress is also involved in the development of obesity and T2D and occurs in liver, pancreas and adipose tissue [30,31,32]. Accumulation of misfolded proteins in the ER lumen, referred as ER stress, occurs when the protein-folding machinery is overwhelmed [33]. ER stress triggers a cellular response, called the unfolded protein response (UPR), to restore ER homeostasis or to induce cell death if the stress is chronic or substantial. Initiation of the UPR arises from the detection of misfolded protein accumulation through three sensor proteins, inositol-requiring enzyme 1α (IRE1α), PKR-like ER kinase (PERK) and activating transcription factor 6 (ATF6), that ultimately activate signaling cascades [33]. Among the signaling pathways activated by the UPR, the integrated stress response (ISR) is highly important in ER stress resolution. ISR is initiated following the phosphorylation of the eukaryotic initiation factor 2 alpha (eIF2α) by PERK. Phosphorylation of eIF2α regulates eIF2 complex activity, leading to the expression of transcription factors such as ATF4, ATF3 and CHOP. Together, these transcription factors regulate the expression of various genes critical to resolve ER stress. Interestingly, ISR and the transcription factor CHOP contribute to obesity and IR [34,35]. Indeed, CHOP-deficient mice are more susceptible to obesity [34], but the molecular mechanism underlying the protective role of CHOP remains unclear.

In the present work, we sought for potential pathophysiological activators of *GDF15* expression in macrophages and identified saturated fatty acids (SFAs) as strong inducers of *GDF15* expression and secretion. We show that SFAs induce *GDF15* expression following ER stress induction. SFA-induced ER stress initiates the activation of the eIF2 complex in a PERK-dependent manner and the expression of the transcription factor CHOP, which directly binds to the *GDF15* promoter region and regulates *GDF15* expression. Interestingly, we show that unsaturated fatty acids (UFAs) prevent SFA-induced *GDF15* expression and ER stress.

## 2. Materials and Methods

### 2.1. Preparation of FFA Solutions

FFA solutions were prepared as previously described [36]. In summary, C16:0 (#P0500, Sigma, St Louis, USA), C16:1 (#P9417, Sigma, St Louis, USA), C18:0 (#S4751, Sigma, St Louis, USA), C18:1 (#O1008, Sigma, St Louis, USA) and C18:2 (#L1376, Sigma, St Louis, USA) were first dissolved in NaOH and then complexed with fatty acid free, low endotoxin BSA (#A8806, Sigma, St Louis, USA) at a FFA:BSA molar ratio of 3.4:1. 

### 2.2. Cell Culture and Treatments

Peripheral blood mononuclear cells (PBMCs) were purified by single step density gradient centrifugation with Ficoll-Paque PLUS (GE Healthcare, Chicago, USA) from buffy coat obtained from healthy donors after informed consent (Croix Rouge de Belgique and Etablissement Francais du Sang). Monocytes were isolated from PBMCs using CD14 MicroBeads (Miltenyi Biotec, Bergisch Gladbach, Germany) according to the manufacturer’s instructions. Monocyte-derived macrophages (MDMs) were generated by culturing freshly isolated monocytes in RPMI 1640 (Lonza, Basel, Switzerland) with 20% heat-inactivated FBS (Gibco, Waltham, USA), 100 IU/mL penicillin (Lonza, Basel, Switzerland), 100 IU/mL streptomycin (Lonza, Basel, Switzerland) and 100 ng/mL of human M-CSF premium grade (Miltenyi Biotec, Bergisch Gladbach, Germany) for 7 days at 37 °C under 5% CO_2_ atmosphere before stimulation. 

The THP-1 monocytic cell lines (ATCC, Manassas, USA) were maintained between 0.5 and 2.0 × 10^6^ cells/mL in RPMI 1640 (Lonza, Basel, Switzerland) supplemented with 10% heat-inactivated FBS (Gibco, Waltham, USA) and 25 µg/mL gentamycin (Gibco, Waltham, USA) at 37 °C under 5% CO_2_ atmosphere. THP-1 cells were differentiated into macrophage-like cells with 100 ng/mL of PMA (#P8139, Sigma, St Louis, USA) for 24 h, washed with PBS, and kept resting for one night in fresh supplemented medium before stimulation.

MDMs and PMA-differentiated THP-1 cells were treated with 100 to 200 µM FFAs or 5 µg/mL tunicamycin (#T7765, Sigma, St Louis, USA) for 2 to 16 h. Inhibitors, including pifithrin-α (#BML-GR325, 20 µM, Enzo Life Sciences, Farmingdale, USA), 4-phenylbutyric acid (PBA) (#P21005, 1 mM, Sigma, St Louis, USA), GSK2606414 (#516535, 2 µM, Sigma, St Louis, USA), GSK2656157 (#sc-490341, 2 µM, Santa Cruz, Dallas, USA), trans-ISRIB (#5284, 1 µM, Tocris, Bristol, UK) and salubrinal (#2347, 50 µM, Tocris, Bristol, UK), were added 15 min before FFAs stimulation and maintained through the experiment. Cell viability for ER stress inhibitors and FFAs was assessed and non-toxic concentrations were selected (Appendix A).

### 2.3. siRNA Transfection

2.5 × 10^5^ THP-1 cells were transfected by using the HiPerFect Transfection reagent (#301705, Qiagen, Venlo, Netherlands) according to manufacturer’s instructions (protocol for suspension cell lines). Predesigned siRNA targeting human IRE1α, PERK, ATF6, CHOP and ATF3 were purchased from Integrated DNA Technologies (IDT, Coralville, USA) (TriFECTa DsiRNA Kit) (Appendix A). For IRE1α and PERK, THP-1 cells were transfected before PMA differentiation and the treatment with FFAs performed 80 h post-transfection. For ATF6, PMA-differentiated THP-1 cells were transfected for 32 h before FFAs stimulation. CHOP and ATF3 siRNAs were transfected in PMA-differentiated THP-1 cells for 8 h before FFAs treatment.

### 2.4. Cell Viability Assay

Cell proliferation reagent WST-1 (#05015944001, Roche Applied Science, Penzberg, Germany) was used to assess cell proliferation, viability and toxicity according to the manufacturer’s instructions. 1.5 × 10^5^ THP-1 cells were differentiated with PMA in 96-well plate followed by treatment for 16 h before addition of WST-1 reagent. 

### 2.5. Luminex Assay

GDF-15 was measured in culture supernatants from MDMs by Luminex assay (R&D Systems, Minneapolis, USA) according to manufacturer’s instructions. Beads were read on a Bio-Plex 200 system (Bio-Rad, Hercules, USA) or on a Bio-Plex MAGPIX system (Bio-Rad, Hercules, USA).

### 2.6. RT-qPCR Analysis

Total RNAs were extracted with high pure RNA isolation kit (Roche Applied Science, Penzberg, Germany) or TRIzol reagent (Ambion, Waltham, USA) according to the manufacturer’s recommendations. DNase treatment was performed on column for RNA isolation kit or after resuspension of RNA pellet for TRIzol extraction by using Dnase I (#EN0521, Thermo Scientific, Waltham, USA). Purified RNAs were reverse-transcribed to complementary DNA (cDNA) by using the high capacity cDNA reverse transcription kit (#4368813, Applied Biosystems, Waltham, USA). qPCR was performed by using Brilliant II SYBR Green QPCR Master Mix (#600828, Agilent, Santa Clara, USA) and ran on a Mx3000P qPCR system (Agilent, Santa Clara, USA) or on a LightCycler 480 (Roche Applied Science, Penzberg, Germany). Gene expressions were calculated using the 2^-ΔΔCT^ method. OAZ1, a highly stable gene in human [37], was chosen as housekeeping gene. Primers were designed with Primer-BLAST (National Center for Biotechnology Information (NCBI)) to amplify all the isoforms of the target gene, except for XBP1-s primers that were designed to only amplify the spliced isoform (isoform 2, NM_001079539). Primer sequences are provided in Appendix A.

### 2.7. Western Blot Analysis

Cells were lysed in total phospholysis buffer (62.5 mM Tris-HCl at pH 6.8, 10% glycerol, 2% SDS, 3% β-mercaptoethanol, 0.03% bromophenol blue, 1 mM DTT, 1 mM sodium orthovanadate, 25 mM β-glycerophosphate, 15 mM sodium fluoride, 1 mM PMSF, and complete protease inhibitor cocktail) and subjected to SDS-PAGE. The following primary antibodies were used: anti-phospho-eIF2α (#9721, Cell Signaling, Danvers, USA; RRID:AB_330951; dilution 1:1000), anti-eIF2α (#9722, Cell Signaling, Danvers, USA; RRID:AB_2230924;dilution 1:1000), anti-CHOP (#sc-575, Santa Cruz, Dallas, USA; RRID:AB_631365; dilution 1:200), anti-HSP90 (#sc-7947, Santa Cruz, Dallas, USA; RRID:AB_2121235; dilution 1:200), anti-HSP60 (#ADI-SPA-806, Enzo Life Sciences, Farmingdale, USA; RRID:AB_10617232; dilution 1:1000), anti-BIP (#sc-13539, Santa Cruz, Dallas, USA; RRID:AB_627698; dilution 1:200) and anti-β-tubulin (#T5201, Sigma, St Louis, USA; RRID:AB_609915; dilution 1:5000). The secondary antibodies used for the revelation were Alexa Fluor 680 anti-mouse IgG (#715-625-150, Jackson ImmunoResearch, West Grove, USA; RRID:AB_2340868; dilution 1:5000), Alexa Fluor 790 anti-rabbit IgG (#711-655-152, Jackson ImmunoResearch, West Grove, USA; RRID:AB_2340628; dilution 1:5000), HRP-linked anti-rabbit IgG (#7074, Cell Signaling, Danvers, USA; RRID:AB_2099233; dilution 1:2000) and HRP-linked anti-mouse IgG (#P0447, Dako, Santa Clara, USA; RRID:AB_2617137; dilution 1:2000). Fluorescent immunoblots were scanned with an OdysseyCLx Imaging System (LI-COR, Lincoln, USA) and quantified with Image studio lite software (version 4.0, LI-COR, Lincoln, USA). For chemoluminescent western blot, revelation was performed with ECL (Pierce, Waltham, USA) by using the digital imaging system ImageQuant LAS 4000 (GE Healthcare, Chicago, USA) and quantification achieved with the ImageQuant TL software (version 7.0, GE Healthcare, Chicago, USA). After phospho-eIF2α revelation, a stripping step of 1 h at 50 °C with Tris-HCl pH 6.8, 2% SDS, 0.8% β-mercaptoethanol was performed before the detection of eIF2α.

### 2.8. ChIP-qPCR Analysis

After 16 h of treatment with BSA or C18:0, PMA-differentiated THP-1 cells were washed twice with PBS and crosslinked with 1% methanol-free formaldehyde (#28908, Pierce, Waltham, USA) for 10 min at room temperature. Crosslink was quenched by addition of glycine. Cells were scrapped in buffer 1 (50 mM HEPES-KOH pH 7.5, 0.25% Triton X100, 10% glycerol, 0.5 mM EGTA, 1 mM EDTA, 140 mM NaCl, 0.5% Igepal, 1 mM PMSF, cOmplete protease inhibitor cocktail) and incubated for 10 min at 4 °C under agitation. Samples were centrifuged at 600g, washed with buffer 2 (10 mM Tris-HCl pH 8.0, 0.5 mM EGTA, 1 mM EDTA, 200 mM NaCl, 1 mM PMSF, cOmplete protease inhibitor cocktail) and centrifuged at 600g. Nuclei pellets were resuspended in buffer 3 (50 mM Tris-HCl pH 8.0, 10 mM EDTA, 1% SDS, 1 mM PMSF, cOmplete protease inhibitor cocktail), sonicated by using a Bioruptor Plus (Diagenode, Seraing, Belgium) and incubated with 2 µg antibody overnight at 4 °C under agitation. Immunocomplexes were captured with protein A/G magnetic beads previously blocked with BSA and tRNA from yeast (#R5636, Sigma, St Louis, USA). Beads were successively washed with buffer 4 (50 mM Tris-HCl pH 8.0, 1% Triton X100, 1 mM EDTA, 150 mM NaCl, 0.1% SDS), buffer 5 (50 mM Tris-HCl pH 8.0, 1% Triton X100, 1 mM EDTA, 500 mM NaCl, 0.1% SDS), buffer 6 (50 mM Tris-HCl pH 8.0, 1 mM EDTA, 500 mM LiCl, 0.5% sodium deoxycholate, 1% igepal) and TE (50 mM Tris-HCl pH 8.0, 1 mM EDTA). Beads were resuspended in elution buffer (1% SDS, 100 mM NaHCO_3_) and incubated overnight at 65 °C with agitation to reverse crosslink. DNA was purified with NucleoSpin gel and PCR Clean-up (#740609, Macherey-Nagel, Düren, Germany) according to the manufacturer’s instructions. qPCR was performed by using Brilliant II SYBR Green QPCR Master Mix (#600828, Agilent, Santa Clara, USA) and ran on a Mx3000P qPCR system (Agilent, Santa Clara, USA). The antibodies used were anti-CHOP (#sc-575, Santa Cruz, Dallas, USA; RRID:AB_631365) and normal rabbit IgG (#2729, Cell Signaling, Danvers, USA; RRID:AB_1031062). Primer sequences are provided in Appendix A.

### 2.9. Statistical Analyses

All statistical analyses were carried out using GraphPad Prism 7 for Windows (GraphPad Software, Inc., San Diego, USA) and presented as the means ± standard error of the mean (SEM). When one independent variable was involved, two-tailed Student’s t-test was performed to compare two groups and one-way ANOVA with Dunnett’s multiple comparisons test to compare more than two groups. Two-way ANOVA with Sidak’s multiple comparisons test was performed when the experiment involved two independent variables. The statistical test used and the number of biological replicates (*n*) are described in each figure legend. Only independent biological replicates, MDM from different donors or independent cell line experiments, were used to draw graph. Biological replicates were obtained from a single measurement, such as western blot, or from the mean of multiple measurements (technical replicates), such as gene expression by RT-qPCR that was assessed in duplicate or cell viability assay in triplicate. Paired statistical tests were used. Statistical significance was set at *p* < 0.05.

## 3. Results

### 3.1. SFAs Promote GDF15 Expression and Secretion in Macrophages

To better understand how *GDF15* expression is regulated in macrophages, we analyzed a publicly available transcriptomic dataset of human primary macrophages treated with several pathophysiological stimuli including cytokines, TLR ligands, glucocorticoids, prostaglandins, lipoproteins and free fatty acids (FFAs) (Figure 1A) [38]. FFAs, and particularly palmitate (C16:0) and stearate (C18:0), the two most common saturated fatty acids (SFAs), induced the highest *GDF15* expression. Since both FFAs and GDF-15 concentrations increase in metabolic diseases [1,2,39], FFAs may represent potential activators of *GDF15* expression. To confirm this finding, we treated human primary monocyte-derived macrophages (MDMs) with physiological concentrations of FFAs (100 µM) [40], including C16:0, C18:0 and their chain length-matched unsaturated fatty acids (UFAs) C16:1, C18:1 and C18:2. We confirmed that SFAs induce *GDF15* expression (Figure 1B) and secretion (Figure 1C) compared to the vehicle alone (BSA) in MDMs. C18:0 induced a stronger response than C16:0. Interestingly, UFAs were unable to induce *GDF15* expression and secretion. 

### 3.2. ER Stress, but Not p53, Is Involved in SFA-Induced GDF15 Expression

GDF-15 was extensively studied for its role in cancer and shown to be mainly regulated by the transcription factor p53 [10] and the ER stress pathway [29]. SFAs did not induce common p53 target genes such as *CDKN1A*, encoding p21, nor *MDM2* (Figure 2A,B), but strongly increased the expression of ER stress-related markers such as *HSP5A*, encoding BIP, and the spliced form of *XBP1* mRNA in MDMs (Figure 2C,D).

Inhibition of p53 with pifithrin-α did not reverse C18:0-induced *GDF15* expression in MDMs (Figure 2E), while ER stress inhibition by the chemical chaperone 4-phenylbutyric acid (PBA) (Appendix A) decreased *GDF15* expression (Figure 2F) and secretion induced by C18:0 (Figure 2G). On the other hand, ER stress induction with tunicamycin, a glycosylation inhibitor leading to the accumulation of misfolded proteins in ER, increased ER stress marker (Figure 2H) and *GDF15* expression (Figure 2I), confirming that ER stress is able to induce *GDF15* expression in macrophages. 

### 3.3. PERK/eIF2/CHOP Pathway Regulates SFA-Induced GDF15 Expression

To better understand how ER stress leads to *GDF15* expression, we inhibited each of the three UPR branches by silencing IRE1α, ATF6 or PERK through siRNA transfection in PMA-differentiated THP-1 cells. The human monocytic cell line THP-1, differentiated in macrophage-like cells by using PMA, was used for its better transfection efficiency compared to primary macrophages. SFAs induced *GDF15* expression and ER stress in PMA-differentiated THP-1 cells as observed in primary MDMs (Appendix A). While the expression of each ER stress sensor was significantly downregulated after transfection of the corresponding siRNA (Appendix A), only PERK silencing decreased C18:0-induced *GDF15* expression (Figure 3A and Appendix A). IRE1α and ATF6 silencing did not decrease *GDF15* expression but significantly reduced their downstream targets (Appendix A). Moreover, pharmacological inhibition of PERK with GSK2606414 or GSK2656157 reduced C18:0-induced *GDF15* expression in PMA-differentiated THP-1 cells (Figure 3B). PERK inhibition also decreased GDF15 expression and secretion following stimulation with C18:0 in primary MDMs (Figure 3C,D). In addition to C18:0-induced *GDF15* expression, tunicamycin-induced *GDF15* expression was also prevented by PERK inhibitors (Figure 3E), demonstrating that the PERK pathway is involved in ER stress-induced *GDF15* expression in macrophages.

As a consequence of ER stress, PERK phosphorylates eIF2α that activates the ISR and increases the expression of *ATF4*, *ATF3* and *DDIT3*, the latter encoding CHOP [33]. In agreement with our results showing that SFAs induce an ER stress response, SFAs increased eIF2α phosphorylation (Appendix A) and expression of *ATF4*, *ATF3* and *DDIT3* (Appendix A). Treatment with trans-ISRIB, an eIF2 complex inhibitor, decreased both C18:0- and tunicamycin-induced *GDF15* expression (Figure 3F). In basal conditions and under ER stress, eIF2α is dephosphorylated by PP1/CReP and PP1/GADD34 complexes driving a negative regulation of eIF2 activity. Interestingly and as anticipated, preventing dephosphorylation of eIF2α by inhibiting these phosphatase complexes with salubrinal exacerbated C18:0- and tunicamycin-induced *GDF15* expression (Figure 3G). Altogether, these results show that the eIF2 signaling pathway, namely the ISR, regulates *GDF15* expression in macrophages.

ATF3 and CHOP are two key transcription factors involved in ISR which are both transcriptionally induced by SFAs (Appendix A). To restrain ATF3 and CHOP induction, we transfected siRNA for 8 h before treatment to prevent the increase of *ATF3* or *DDIT3* mRNA level following C18:0 treatment (Appendix A). CHOP silencing decreased *GDF15* induction by C18:0 (Figure 4A), while ATF3 knock down did not (Figure 4B). Analysis of the *GDF15* promoter region revealed seven potential CHOP response elements that we grouped in 4 sites (Figure 4C). To determine whether CHOP is recruited to the *GDF15* promoter in response to C18:0 treatment, we performed a ChIP-qPCR assay. BSA treatment alone did not enriched CHOP at any sites suggesting no or few basal bindings of CHOP to the *GDF15* promoter (Figure 4D), an observation corroborated by the undetectable CHOP protein levels under basal conditions (Figure 4E). C18:0 increased CHOP protein levels (Figure 4E) and CHOP binding to the *GDF15* promoter (Figure 4D). These results suggest that CHOP regulates *GDF15* expression in macrophages by direct binding to its promoter. It is worth noting that all the previous treatments restraining *GDF15* expression such as PBA, PERK siRNAs, PERK inhibitors and trans-ISRIB also decreased *DDIT3* expression (Figure 5A–E), while treatments exacerbating *GDF15* expression, such as with the p53 inhibitor pifithrin-α or salubrinal, increased *DDIT3* expression (Figure 5F,G). Finally, IRE1α and ATF6 silencing by siRNA, which did not modulate *GDF15* expression, had no effect on *DDIT3* expression (Appendix A). Collectively, these data demonstrate that C18:0 promotes *GDF15* expression in macrophages following the induction of an ER stress through the PERK/eIF2/CHOP signaling pathway.

### 3.4. UFAs Inhibit SFA-Induced ER Stress and GDF15 Expression

Unlike SFAs, UFAs did not induce *GDF15* expression nor secretion (Figure 1B,C). It has been previously reported that UFAs can prevent SFA-induced cytokine production such as IL-1β or IL-6 [36,41,42]. To investigate whether UFAs prevent C18:0-induced *GDF15* expression, PMA-differentiated THP-1 cells were treated with C18:0 alone or in combination with equimolar concentrations of UFAs. Co-treatments with C16:1, C18:1 and C18:2 prevented C18:0-induced *GDF15* expression (Figure 6A). Since CHOP is involved in SFA-induced *GDF15* expression, the ER stress response upon co-treatment with C18:0 and UFAs was analyzed. UFA treatments restrained C18:0-induced ER stress markers including the splicing of *XBP1* mRNA (Figure 6B), eIF2α phosphorylation (Figure 6C) and the expression of *HSPA5* and *DDIT3* (Figure 6D,E). In contrast, tunicamycin-induced GDF15 expression (Figure 6F) and ER stress (Figure 6G–J) were not inhibited by UFAs co-treatment, suggesting that UFAs have no general effect on ER stress but specifically restrain the effect of SFAs on ER stress. The protection conferred by UFA treatment against C18:0-induced ER stress and CHOP expression likely account for the inhibition of *GDF15* expression. Taken together, our results show the complex action of FFAs on *GDF15* regulation in macrophages with SFAs promoting GDF-15 production and UFAs counteracting SFAs effect. 

## 4. Discussion

The role of GDF-15 and its receptor GFRAL in the development and progression of obesity and T2D is well described [12,13,14,16,17,18,19,20], but the molecular mechanisms accounting for the high expression of GDF-15 in these conditions is poorly understood. Obesity increases *GDF15* expression in liver and adipose tissue [28]. Parenchymal cells from these tissues (i.e., hepatocytes, preadipocytes and mature adipocytes) have been proposed as sources for the increased blood levels [22,23,28]. Macrophages are the main immune cell population in both adipose tissue and liver and expand with obesity [26]. Recently, it was shown that *Gdf15* deletion in macrophages induces a more severe obesity and IR upon high fat diet feeding [15], suggesting that macrophages also participate to the obesity-associated increase of GDF-15.

To better understand how *GDF15* expression may be regulated in macrophages, we investigated the effect of several physiological stimuli and identified SFAs as promoters of *GDF15* expression and secretion. C18:0 (stearate) induced a stronger *GDF15* expression and secretion than C16:0 (palmitate) at the same concentration (Figure 1B,C). C18:0 is the second most important SFA, representing 12.5% of plasma FFAs, while C16:0 represents 28% [40]. Several studies reported a positive association for both C16:0 and C18:0 with obesity and T2D [43,44].

We showed that C18:0-induced *GDF15* expression is ER stress-dependent and involves the UPR/ISR via the PERK/eIF2/CHOP signaling pathway and direct binding of the transcription factor CHOP to the *GDF15* promoter. In basal conditions, macrophages express no or low levels of CHOP (Figure 4E), likely explaining the absence of CHOP binding to the GDF15 promoter (Figure 4D). Moreover, ER stress, UPR and ISR inhibitors have no effect in untreated cells, suggesting that basal *GDF15* expression and secretion in macrophages is ER stress-independent. However, upon SFA treatment, the PERK/eIF2/CHOP signaling pathway is activated leading to CHOP expression and binding to the *GDF15* promoter, *GDF15* expression and secretion. The role of ER stress and ISR on *GDF15* regulation was previously described in cancer cells [29], myocytes [13], gut epithelial cells [24] and hepatocytes [23]. These studies revealed that various ISR stimuli, including the non-steroidal anti-inflammatory compound sulindac sulfide [29], mitochondrial UPR [13], enteropathogenic *Escherichia coli* [24], tunicamycin [23,28] and thapsigargin [23,28], induce GDF-15. In addition to those different ER stress stimuli known to promote *GDF15* expression, SFAs represent interesting pathophysiological stimuli linking obesity and GDF-15. Indeed, obesity and IR increase FFA levels and a disequilibrium in the SFA/UFA balance is commonly associated with progression of obesity and the development of complications such as T2D [43,44]. The transcription factor CHOP has also been involved in obesity development and CHOP-deficient mice are more susceptible to obesity [34], similarly to *Gdf15*-deficient mice [18]. The molecular mechanism accounting for the higher weight gain of CHOP-deficient mice is not elucidated but, given the role of CHOP in the regulation of *GDF15* and the role of GDF-15 in the control of food intake, a decrease of GDF-15 level in these mice may play a role.

Previous studies have demonstrated that UFAs can prevent cytokines production by SFAs, such as IL-6 [42] and IL-1β [36,41]. We observed that equimolar concentrations of UFAs also prevent C18:0-induced *GDF15* expression in macrophages (Figure 6A). UFAs also suppress ER stress and CHOP expression, probably accounting for the inhibition of *GDF15* expression. Inhibition of ER stress by UFAs is specific to SFAs as UFAs were unable to protect against tunicamycin-induced ER stress and *GDF15* expression. These results reveal the complexity of FFA-mediated regulation of inflammatory process and provide evidences than the ratio SFAs/UFAs may be more important than the absolute value of SFAs.

GDF-15 is protective against obesity and T2D [12,13,14,16,17,18,19,20]. However, GDF-15 shows a positive correlation with obesity progression and the development of complications such as T2D [1,2,3,4]. This puzzling discrepancy may result from SFAs properties. In vitro, SFAs activate macrophages to produce pro-inflammatory cytokines such as TNF-α and IL-1β [36,45]. In obese adipose tissue, macrophages develop a pro-inflammatory phenotype in line with the presence of metabolic stimuli such as SFAs [46]. Since SFAs induce a pro-inflammatory phenotype in macrophages [45], GDF-15 may be produced besides other cytokines and be considered as an independent marker of macrophage activation by SFAs. The positive association between GDF-15 level and obesity progression may therefore result from the pro-inflammatory status of SFA-activated macrophages, given that pro-inflammatory macrophages are involved in the development of obesity complications [26]. In conclusion, we show that SFAs promote *GDF15* expression and secretion in macrophages. SFA-induced *GDF15* expression involves ER stress and the PERK/eIF2/CHOP signaling pathway. UFAs prevent SFA-induced *GDF15* expression and ER stress. These new findings provide new molecular insights about how diet and lipid metabolism may regulate the development of obesity and T2D, but also suggest that GDF-15 may be a marker of the metabolic activation of macrophages during obesity.

## Figures and Tables

**Figure 1 nutrients-12-03771-f001:**
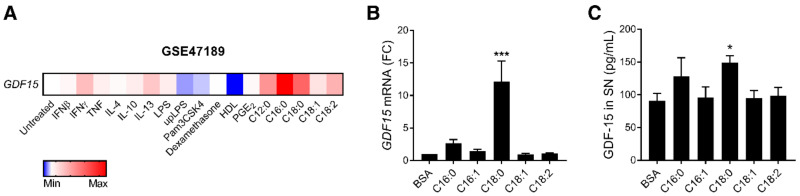
Saturated fatty acids (SFAs) promote *GDF15* expression and secretion in macrophages. (**A**) Heatmap representing *GDF15* expression in human primary macrophages treated with different stimuli. Further details are available in Xue et al., 2014 [38] and under the accession number GSE47189. (**B**,**C**) MDMs were treated with 100 µM FFA or BSA for 16 h. *GDF15* expression was assessed by RT-qPCR (**B**) and GDF-15 secretion in SN was measured by luminex assay (**C**) (*n* = 5). * *p* < 0.05; *** *p* < 0.001 by one-way ANOVA with Dunnett’s multiple comparisons test. Results are presented as mean ± SEM. SN, supernatant. FC, fold change.

**Figure 2 nutrients-12-03771-f002:**
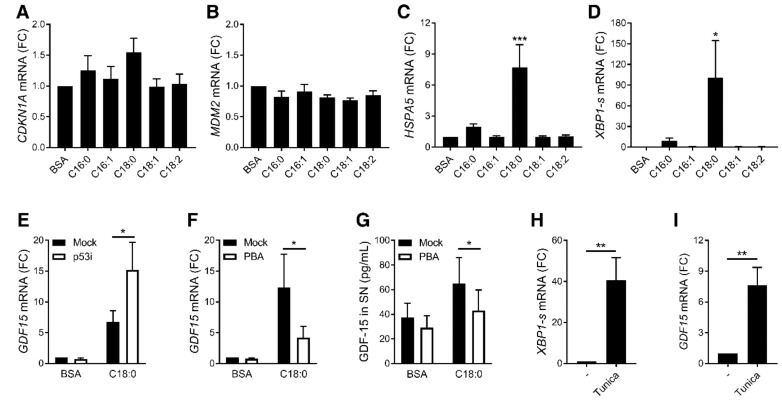
Role of ER stress in SFA-induced *GDF15* expression. (**A**–**D**) mRNA expression measured by RT-qPCR after treatment with 100 µM FFAs for 16 h in MDMs (*n* = 5). * *p* < 0.05; *** *p* < 0.001 by one-way ANOVA with Dunnett’s multiple comparisons test. (**E**–**G**) *GDF15* expression measured by RT-qPCR (**E**,**F**) and GDF-15 secretion in SN measured by luminex assay (**G**) after stimulation of MDMs with 100 µM C18:0 for 16 h in presence of 20 µM pifithrin-α (p53i) (*n* = 3) or 1 mM PBA (*n* = 5). * *p* < 0.05 by two-way ANOVA with Sidak’s multiple comparisons test. (**H**,**I**) mRNA expression measured by RT-qPCR after treatment with 5 µg/mL tunicamycin for 16 h in MDMs (*n* = 5). ** *p* < 0.01 by Student’s *t*-test. Results are presented as mean ± SEM. SN, supernatant. FC, Fold change.

**Figure 3 nutrients-12-03771-f003:**
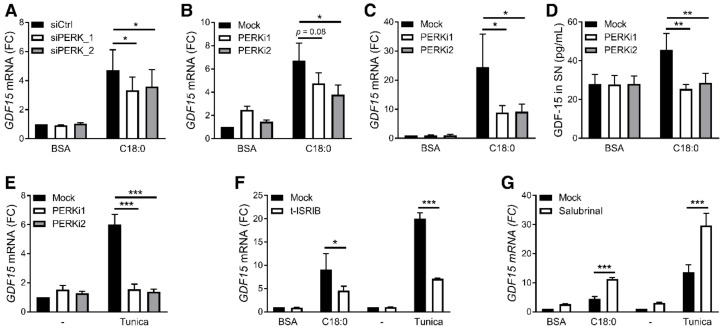
Role of ISR in SFA-induced GDF15 expression. (**A**,**B**) *GDF15* expression measured by RT-qPCR in PMA-differentiated THP-1 cells treated with 200 µM C18:0 for 16 h after PERK silencing by siRNA (**A**) (*n* = 5) or in presence of 2 µM GSK2606414 (PERKi1) or GSK2656157 (PERKi2) (**B**) (*n* = 5). (**C**–**E**) *GDF15* expression measured by RT-qPCR (**C**,**E**) or GDF-15 secretion in supernatant (SN) measured by luminex assay (**D**) in MDMs treated with 100 µM C18:0 or 5 µg/mL tunicamycin for 16 h in presence of 2 µM GSK2606414 (PERKi1) or GSK2656157 (PERKi2) (*n* = 4). (**F**,**G**) *GDF15* expression measured by RT-qPCR after treatment with 200 µM C18:0 or 5 µg/mL tunicamycin for 16 hrs in PMA-differentiated THP-1 cells in presence of 1 µM trans-ISRIB or 50 µM salubrinal (*n* = 5). * *p* < 0.05, ** *p* < 0.01, *** *p* < 0.001 by two-way ANOVA with Sidak’s multiple comparisons test. Results are presented as mean ± SEM. FC, fold change.

**Figure 4 nutrients-12-03771-f004:**
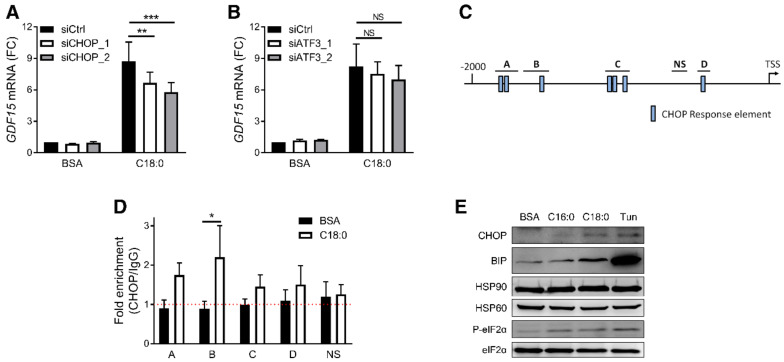
Role of CHOP in SFA-induced *GDF15* expression. (**A**,**B**) *GDF15* expression measured by RT-qPCR after treatment with 200 µM C18:0 for 16 h in PMA-differentiated THP-1 after ATF3 or CHOP silencing by siRNA (*n* = 4). (**C**) Schematic representation of *GDF15* promoter with the putative CHOP response elements and the regions analyzed in ChIP-qPCR. (**D**) CHOP binding in *GDF15* promoter measured by ChIP-qPCR in PMA-differentiated THP-1 cells treated with 200 µM C18:0 for 16 h (*n* = 4). (**E**) Analysis of CHOP, BIP and eIF2α phosphorylation by western blot after stimulation of PMA-differentiated THP-1 cells with 200 µM SFAs or 5 µg/mL tunicamycin for 24 h (*n* = 3). HSP90, HSP60 and eIF2α were used as loading control.* *p* < 0.05, ** *p* < 0.01, *** *p* < 0.001 by two-way ANOVA with Sidak’s multiple comparisons test. Results are presented as mean ± SEM. FC, Fold change. TSS, transcription start site.

**Figure 5 nutrients-12-03771-f005:**
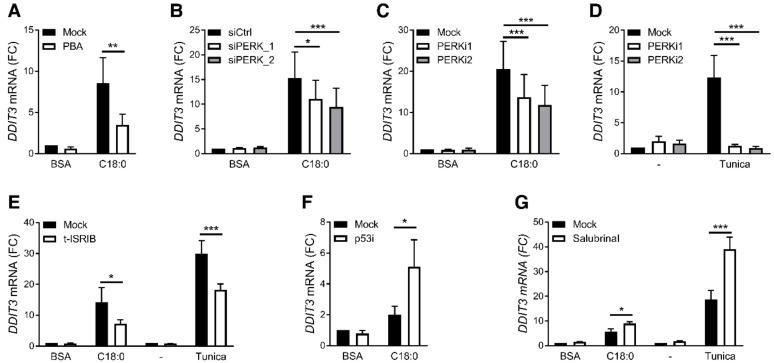
CHOP regulation by pharmacological and siRNA strategies. (**A**–**G**) *DDIT3* expression measured by RT-qPCR after treatment with 100 µM C18:0 in MDMs (**A**,**C**,**D**,**F**), 200 µM C18:0 in PMA-differentiated THP-1 (**B**,**E**,**G**) or 5 µg/mL tunicamycin for 16 h in presence of 1 mM PBA, 2 µM GSK2606414 (PERKi1), 2 µM GSK2656157 (PERKi2), 1 µM trans-ISRIB, 20 µM pifithrin-α (p53i), 50 µM salubrinal or after PERK silencing by siRNA (*n* = 3–5). * *p* < 0.05, ** *p* < 0.01, *** *p* < 0.001 by two-way ANOVA with Sidak’s multiple comparisons test. Results are presented as mean ± SEM. FC, Fold change.

**Figure 6 nutrients-12-03771-f006:**
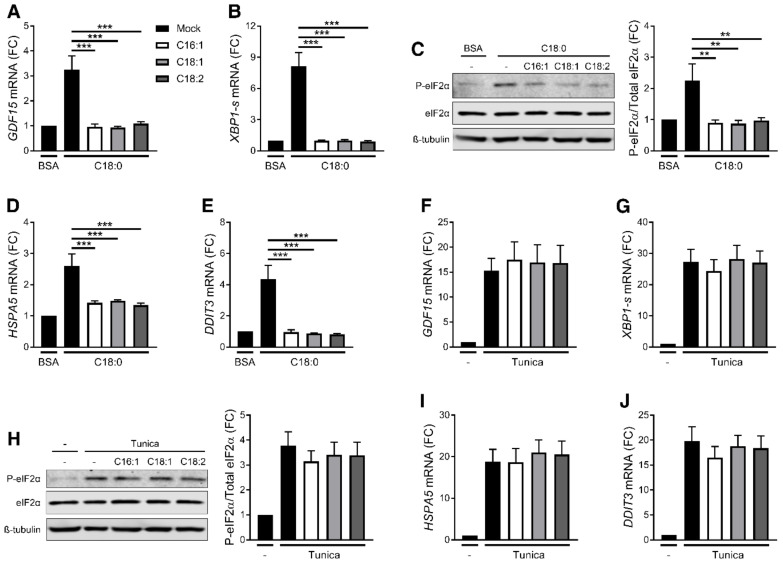
UFAs prevent SFA-induced *GDF15* expression. (**A**–**J**) mRNA expression measured by RT-qPCR (**A**,**B**,**D**–**G**,**I**,**J**) and analysis of eIF2α phosphorylation by western blot (**C**,**H**) after treatment with 200 µM C18:0 or 5 µg/mL tunicamycin in presence of 200 µM UFAs for 16 h in PMA-differentiated THP-1 cells (*n* = 5). ** *p* < 0.01, *** *p* < 0.001 by one-way ANOVA with Dunnett’s multiple comparisons test. Results are presented as mean ± SEM. FC, Fold change.

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
