# Peer review of "Saturated Fatty Acids Promote GDF15 Expression in Human Macrophages through the PERK/eIF2/CHOP Signaling Pathway"

_nutrients, 2020, doi:10.3390/nu12123771_

Round 1
Reviewer 1 Report
The authors aimed to evaluate the effect of saturated fatty acids on the expression of GDF15 with a special emphasis on the molecular mechanism underlined by PERK/eIF2/CHOP pathway. The figures are well organized, the rationale is solid and the experimental plan is logical.
Unfortunately, several flaws were identified which deserve the authors’ attention. These are listed below:
- Please provide details regarding antibodies: dilutions, RRIDs.
- Please provide details on statistical analyses. How many biological and technical repetitions were carried out?
- Please provide evidence on the inhibition of p53 with pifithrin-a and inhibition of ER stress by PBA
- Where the levels of unspliced XBPs measured?
- For all XBPs measurements please provide standard PCR bands showing the ratio of unspliced/spliced XBPs.
- Please provide Western Blot analyses confirming GDF15 levels in macrophages treated with SFAs.
- Please provide all details regarding siRNAs used.
- Please provide the whole membranes of WBs. It seems that some of them (especially CHOP) have high background and unspecific signals.
- Please provide marker for all analyzed proteins
- Where the WBs normalized against housekeeping protein?
- Please provide beta-actin WB for figure 4E.
- All figures are of extremely low quality
- The Discussion part should follow in different para in context of results and figure numbers/symbols.
- Also, the discussion is limited to several statements, please elaborate
- How the primer sequences were designed? Please provide accession numbers
- Please provide annealing temps for primers.
- The efficiency of silencing was confirmed with RT-PCRs. Please provide WBs.
- Line 106 and 109 should be CO2
- Line 161 should be β-mercaptoethanol
- Line 179 should be NaHCO3
Reviewer 2 Report
Induction of GDF15 by PERK-CHOP pathway is previously proven. The current manuscript is incremental to the current knowledge. The key finding of this paper is showing this phenomenon in macrophages. Current manuscript need following correction.
- For figure 2 it would be essential to see SFA induce ER stress in macrophages
- For figure 3, the statistics for panel A and B need to evaluated again. With such a big error bar by SEM it is difficult to see statistical significance.
- For figure 3, the readers need to see knockdown by siRNA
Reviewer 3 Report
The authors investigate the effects of saturated FA on GDF15.
Minor grammatical changes in the abstract required eg line 17, 18
Plasmatic? Is there a more appropriate word?
It is not clear the link between obesity, anorexia and GDF-15. The authors write as if obesity = anorexia line 40
GFRAL is not only restricted to the brain https://www.aging-us.com/article/103830/text line 45 (this is also contradicted by line 54+ in the manuscript)
The introduction does not justify the experiments focused on SFA. The introduction discusses GDF-15, and the reader is left confused as to why SFA was used to investigate regulators of GDF-15.
Western-were standard amounts of proteins analysed (loaded on the gel)?
Parametric statistical analysis assumes normally distributed data. Was a test for normality done?
Was OAZ1 variable among the samples?
Figure 3: were Western blots performed to ensure knockout of the protein by siRNA?
The quality of the Western for 4E is poor, one of the bands is chopped in half. Do the authors have a better represented Western blot?
Round 2
Reviewer 1 Report
I still have a huge problem with Western Blot against CHOP (Figure 4E). The marker is almost not visible, there is no positive control and the background is extremely high. I'm not convinced that marked bands correspond to CHOP. Please use another antibody to confirm observations.
Reviewer 2 Report
All of my doubts have been answered.
Author Response
Reviewer 2 was satisfied and had no further comments.
Reviewer 3 Report
Comments have addressed my questions
Author Response
Reviewer 3 was satisfied and had no further comments.